# Structural insights into SorCS2–Nerve Growth Factor complex formation

Nadia Leloup [1], Lucas M.P. Chataigner[1] & Bert J.C. Janssen [1]

Signaling of SorCS receptors by proneurotrophin ligands regulates neuronal plasticity, induces apoptosis and is associated with mental disorders. The detailed structure of SorCS2 and its extracellular specificity are unresolved. Here we report crystal structures of the SorCS2–NGF complex and unliganded SorCS2 ectodomain, revealing cross-braced SorCS2 homodimers with two NGF dimers bound in a 2:4 stoichiometry. Five out of six SorCS2 domains directly contribute to dimer formation and a C-terminal membrane proximal unreported domain, with an RNA recognition motif fold, locks the dimer in an intermolecular head-to-tail interaction. The complex structure shows an altered SorCS2 conformation indicating substantial structural plasticity. Both NGF dimer chains interact exclusively with the top face of a SorCS2 β-propeller. Biophysical experiments reveal that NGF, proNGF, and proBDNF bind at this site on SorCS2. Taken together, our data reveal a structurally flexible SorCS2 receptor that employs the large β-propeller as a ligand binding platform.

[1] Crystal and Structural Chemistry, Bijvoet Center for Biomolecular Research, Faculty of Science, Utrecht University, 3584 CH Utrecht, The Netherlands. Correspondence and requests for materials should be addressed to B.J.C.J. (email: b.j.c.janssen@uu.nl)

The type I transmembrane receptor Sortilin-related CNS-expressed receptor 2 (SorCS2), together with SorCS1 and 3, Sortilin and SorLA constitute the Vacuolar Protein Sorting 10 protein (VPS10p) family that is central to many pathways in control of neuronal viability and function, and has been associated with cancer and neurodegenerative diseases such as Alzheimer's[1] and Huntington's[2]. Two roles have been identified for VPS10p members; in particular, SorCS2 and Sortilin are well studied for their function as extracellular receptors for the cognate proneurotrophin ligands to regulate synaptic plasticity and trigger apoptotic signaling[3–6], and they are responsible for binding and sorting a diverse set of ligands for secretion, internalization and endosome to lysosome sorting[7–9].

The defining feature of the VPS10p family, the extracellular VPS10p subunit, is critical for interactions with a multitude of ligands[4,10–13] of which the proneurotrophin class is the most important. Crystal structures of the VPS10p subunits of Sortilin[14] and SorLA[15] have revealed that the VPS10p subunit consists of a ten-bladed β-propeller followed by two cysteine-rich domains called 10CC-a and 10CC-b, which interact with and stabilize the β-propeller. The central tunnel in the β-propeller contains binding sites for peptide ligands such as neurotensin[14] and Aβ[15], and these peptides bind by extending the β-sheet of a propeller blade. It was recently shown that the Sortilin VPS10p subunit undergoes a conformational change and dimerizes at low pH; it is believed that both the dimerization and conformational change trigger release of a diverse set of ligands, including (pro)neurotrophins, at low pH[16,17]. While the ectodomain of Sortilin consists of the VPS10p subunit only, other members of the family possess additional domains C-terminal of the VPS10p subunit which may regulate signaling differentially from Sortilin[18]. SorLA contains a large low density lipoprotein receptor repeat region and a fibronectin-type III repeat that enable binding and release of apolipoprotein E in a fashion similar to the low-density lipoprotein receptor[15]. The members of the SorCS subfamily (SorCS1, SorCS2, and SorCS3) all contain a region rich in leucine residues that consists of a polycystic kidney disease (PKD) domain (pdb-id 1WGO) and an additional 202 residues of unknown fold. Recent low resolution negative stain electron microscopy structures have shown that all SorCS subfamily members (SorCS1-3) dimerize through the leucine-rich region[19]. But no high-resolution information is available for the VPS10p subunit of any SorCS member and the details of SorCS dimerization are unresolved.

Proneurotrophins and their proteolytic processed mature forms, neurotrophins, have predominantly distinct functions. Proneurotrophins, such as pro-nerve growth factor (proNGF) and pro-brain-derived neurotropic factor (proBDNF), can trigger neuronal apoptosis, growth cone retraction, and regulate neuronal plasticity by forming a ternary complex with VPS10p members SorCS2 or Sortilin and the p75 neurotrophic receptor (p75NTR)[3–6]. Neurotrophins, on the other hand, function as growth factors, and induce growth and survival of neurons by binding the receptors tropomyosin receptor kinase (Trk) and p75NTR[7]. Both proneurotrophins and neurotrophins bind to VPS10p members (such as Sortilin, SorCS2, and SorCS3), p75NTR and Trk, but in general the affinity of proneurotrophin is higher for VPS10p members while that of neurotrophins is higher for p75NTR and Trk[4,6], although there is one exception; NGF binds with higher affinity to SorCS3 than does proNGF[20]. The binding affinity is enhanced substantially when proNGF binds simultaneously to cell-surface expressed SorCS2 (or Sortilin) and p75NTR to form a ternary complex and this ternary complex is required for signaling[3,4,6,21].

How proneurotrophins interact with SorCS2 or other VPS10p members is not well understood but structures of NGF and proNGF in complex with p75NTR[12,22] and of NGF in complex

with Trk[23,24] have revealed how NGF and proNGF homodimers engage the p75NTR and Trk receptors via an overlapping binding site on the mature domain of NGF. The structure of the mature NGF part in NGF and proNGF is identical, except for the repositioning of one loop[12,25]. The pro domain of neurotrophins is largely disordered[26,27] and was not resolved in the crystal structure of the proNGF–p75NTR complex[12].

To resolve the details of the SorCS2 extracellular segment, its homo-dimerization and complex formation with NGF and proNGF we have solved crystal structures of the unliganded SorCS2 ectodomain and the SorCS2–NGF complex, and quantified interactions of wild-type and a structure-guided SorCS2 mutant to NGF, proNGF, and proBDNF. We show that the unliganded SorCS2 full extracellular segment consists of six domains of which two were unreported. Five domains contribute directly to forming a cross-braced dimer. The NGF-bound SorCS2 structure reveals an altered SorCS2 conformation. Two NGF dimers bind to the SorCS2 dimer in a symmetric 2:4 SorCS2:NGF stoichiometry. ProNGF and proBDNF also bind to SorCS2 at the NGF binding site identified in the SorCS2–NGF crystal structure. The binding sites for p75NTR, Trk and SorCS2 on NGF are partially overlapping, but as both NGF and proNGF are dimers and thus contain two equivalent receptor binding sites, it is possible that proNGF can engage cell-surface expressed SorCS2 and p75NTR simultaneously to trigger signaling.

## Results

**The SorCS2 structure reveals a cross-braced homodimer.** We have determined the crystal structure of the unliganded SorCS2 ectodomain at 4.2 Å resolution (see Methods, Fig. 1, Table 1). The structure of the entire mature extracellular segment of mouse SorCS2 (sSorCS2, residues 117–1072, excluding the signal- and pro-peptide), reveals a cross-braced dimer (Fig. 1b). Each chain consists of six domains; a large N-terminal ten-bladed β-propeller followed by two 10CC domains (10CC-a and 10CC-b), two polycystic kidney disease (PKD) domains (PKD1 and PKD2) with an FN-III topology, and an unpredicted C-terminal membrane proximal domain (residues 958-1066) that we termed SorCS membrane proximal (SoMP) (Fig. 1c). The SoMP domain adopts a fold similar to an RNA Recognition Motif (RRM) with a core consisting of three antiparallel β-strands in a β-sheet that is supported by two α-helixes (Fig. 1d). Sequence analysis indicates that SorCS1 and 3 share this domain composition including the SoMP domain (Supplementary Fig. 1). Together the six domains adopt an arc shape with a maximum dimension of 140 Å (Fig. 1c). The two 10CC domains wrap around and interact with the bottom face of the β-propeller, leaving the β-propeller top face exposed (by convention the top face is defined as the β-propeller surface that contains the DA and BC loops[28]). The two PKD domains are pointing away from the β-propeller – 10CC combination in a head-to-tail fashion. The SoMP makes a sharp turn with respect to PKD2 and interacts with the side of this domain. Except for 10CC-a all domains participate in the large dimerization interface that has a buried surface area of 5790 Å$^2$ (Fig. 1e). In the dimer, the two β-propellers are located at the sides, and each β-propeller interacts with the SoMP domain of the other chain. The PKD domains of both chains twist around each other in an anti-parallel fashion in which PKD1 interacts with PKD2 of the other dimer chain and vice versa. The pseudo two-fold homodimer symmetry axis passes through the center of this PKD-based interface and is perpendicular to the predicted cell-surface location (Fig. 1b). The two chains in the dimer adopt slightly different conformations with largest differences in the 10CC-b–PKD1 and in the PKD2–SoMP connection (root mean square deviation (RMSD) of 1.8 Å over 909 out of 915 Cα atoms,

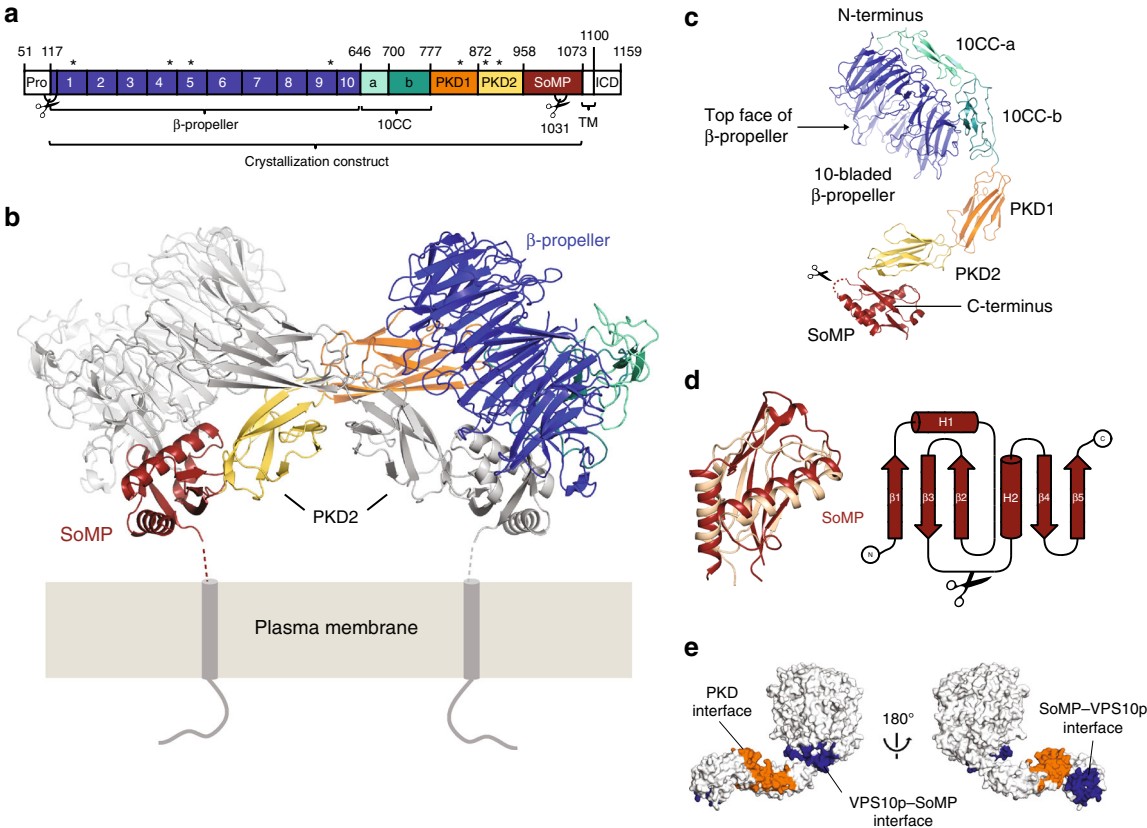

**Fig. 1** SorCS2 forms a cross-braced homodimer. **a** Schematic representation of SorCS2 with the different domains colored and the ten blades of the β-propeller numbered. The second PKD domain and the SoMP domain were previously unreported and together with PKD1 constitute the leucine rich repeat region. Predicted N-glycosylation sites (stars), proteolytic processing sites (scissors), pro domain (PRO), transmembrane domain (TM) and intracellular domain (ICD) are also indicated. **b** Cartoon representation of the structure of the unliganded sSorCS2 dimer (one monomer colored according to (**a**), the other monomer colored gray). The predicted orientation and connection to the cell surface of sSorCS2 are indicated. Seven residues are lacking between the structure and the two transmembrane helixes. The top faces of the β-propellers are slanted towards the cell membrane. The SoMP domain is folded and interacts with the β-propeller of an opposing monomer, essentially cross-bracing the dimer. **c** One chain of the sSorCS2 dimer in cartoon representation and colored as in (**a**). The processing site at S1031[4] (scissors) resides in a disordered loop that is located close to the cell surface. **d** SoMP has a fold similar to an RNA Recognition Motif. Superposition of the SorCS2 SoMP domain (red) and the RNA recognition motif of yeast eIF3b[66] (pdb 3NS5, yellow) (left panel). Topology diagram of SorCS2 SoMP with secondary structure elements indicated (right, panel). Processing site (scissors) as in (**c**). **e** Residues involved in dimer formation colored orange for the PKD–PKD interface and blue for the SoMP–VPS10p interface

see Supplementary Fig. 2). In agreement with recent data from others[19], we find that sSorCS2 also forms a dimer in solution (Supplementary Fig. 3a and b). Furthermore, the crystal structure of the sSorCS2 dimer fits well to the solution state determined by small-angle X-ray scattering (Supplementary Fig. 3c–f).

**The SorCS2 VPS10p resembles that of Sortilin and SorLA.** The family-defining VPS10p subunit of SorCS2 is similar to those of the two other structurally characterized VPS10p subunits of Sortilin[14,16,17,29] and SorLA[15]. The positions of the three domains with respect to each other are equivalent (Supplementary Fig. 4) and the structure of the β-propeller does not reveal major differences between the members. Rmsd of 1.8 Å over 467 Cα atoms are found when comparing the β-propeller of SorCS2 with that of either ligand bound or unliganded SorLA (pdb-id 3WSY and 3WSX, respectively)[15], while rmsds of 1.9 Å over 450 Cα atoms and 2.1 Å over 448 Cα atoms are found when comparing the β-propeller of SorCS2 with that of Sortilin monomer (pdb-id 3F6K)[14] and Sortilin dimer (pdb-id 5NMT)[17], respectively. The structures of the 10CC domains vary more among the VPS10p members, but this may reflect the limited secondary structure, small hydrophobic core and the structural plasticity within these

domains that has been described for 10CC-b of Sortilin and SorLA (Supplementary Fig. 4)[15–17].

**NGF binds to each SorCS2 β-propeller in a 2 to 4 complex.** The crystal structure of NGF bound to sSorCS2 was determined at 3.9 Å resolution (see Methods, Table 1) and shows a symmetric complex, defined by a crystallographic two-fold axis, with a 2:4 (sSorCS2:NGF) stoichiometry (Fig. 2). An NGF homodimer is bound onto the top face of each β-propeller of the sSorCS2 homodimer. Both NGF dimer chains interact exclusively with the sSorCS2 β-propeller in a mixed hydrophobic-hydrophilic interface of 1920 Å² buried surface area that is conserved both among SorCS2 and NGF orthologues and paralogues (Fig. 2b) suggesting that this binding mode is common to SorCS members and proneurotrophins. NGF latches onto the β-propeller top face, similar to a hand holding a tire (Fig. 3). NGF β-strands A and B of one monomer and C and D of the other monomer form the palm of the hand that has predominantly hydrophobic interactions with loops on the top face of β-propeller blades 8, 9 and 10 that are part of the tire. The hand fingers, formed by β-strands A′ and A″ and the loop connecting strands C and D (L4)[23], are positioned above the central β-propeller tunnel and interact with loops on

**Table 1 Data collection and refinement statistics**

|  | sSorCS2 | sSorCS2-NGF |
|---|---|---|
| Data collection |  |  |
| Space group | C222$_1$ | C2 |
| Cell dimensions |  |  |
| $a, b, c$ (Å) | 138.8, 329.3, 131.4 | 229.8, 117.7, 90.0 |
| $\alpha, \beta, \gamma$ (°) | 90, 90, 90 | 90, 111.9, 90 |
| Resolution (Å) | 49.20–4.20 (4.53–4.20)$^a$ | 60.84–3.90 (4.27–3.90) |
| $R_{merge}$ | 0.178 (1.10) | 0.269 (1.07) |
| $I / \sigma I$ | 5.6 (1.5) | 5.8 (1.7) |
| Completeness (%) | 99.8 (99.6) | 99.9 (99.6) |
| Redundancy | 6.3 (6.5) | 6.0 (6.1) |
| $CC_{1/2}$ | 0.996 (0.670) | 0.992 (0.572) |
| Refinement |  |  |
| Resolution (Å) | 49.20–4.20 | 60.68–3.90 |
| No. reflections | 22,379 | 20,410 |
| $R_{work}/R_{free}$ | 0.247/0.291 | 0.257/0.303 |
| No. atoms |  |  |
| Protein | 14,558 | 9030 |
| Ligand/ion | 154 | 81 |
| Water | 0 | 0 |
| B-factors (Å$^2$) |  |  |
| Protein | 196 | 134 |
| Ligand/ion | 228 | 174 |
| Water | n.a. | n.a. |
| RMS deviations |  |  |
| Bond lengths (Å) | 0.002 | 0.003 |
| Bond angles (°) | 0.59 | 0.61 |
| Ramachandran |  |  |
| Favored (%) | 90.7 | 89.1 |
| Outliers (%) | 0.0 | 0.0 |
| Rotamer |  |  |
| Outliers (%) | 1.3 | 1.57 |
| Molprobity score | 1.78 | 2.07 |

$^a$Values in parentheses are for highest-resolution shell.

the top face of blades 1, 6, and 8. Finally, the hand thumb, consisting of the N-terminus of NGF, interacts with loops in blades 9 and 10. Surface Plasmon resonance analyses support the idea that NGF, proNGF and proBDNF interact in this mode with sSorCS2 (Fig. 4a, b and Supplementary Fig. 5). Binding affinities of $236 \pm 78$ nM for NGF ($n = 3$), $43 \pm 12$ nM for proNGF ($n = 3$), and $69 \pm 1$ nM for proBDNF ($n = 2$), are disrupted by introducing an N-linked glycan in the NGF binding site on sSorCS2 (F630N) (Figs. 3d, 4a and Supplementary Fig. 5). In short, in the sSorCS2–NGF complex two NGF dimers each interact with sSorCS2 at one side while the equivalent symmetric receptor binding site in each NGF dimer remains free and exposed.

**Ligand bound and free sSorCS2 adopt different conformations.** SSorCS2 in complex with NGF has a substantially different conformation compared to the unliganded sSorCS2 (r.m.s.d. of 4.8 and 5.5 Å over 909 out of 915 Cα atoms comparing ligand bound sSorCS2 to the two unliganded sSorCS2 chains, Supplementary Movie 1) whereas in NGF only the N-terminus changes position (Fig. 4c, d). The conformational difference in sSorCS2 may be induced by NGF ligand binding or could be a result of intrinsic conformational plasticity in SorCS2. Compared to unliganded sSorCS2 NGF bound sSorCS2 has extended in height from 75 to 86 Å and decreased in width from 179 to 156 Å, the β-propeller has rotated 30 degrees and moved 30 Å away from the predicted location of the cell membrane while its interaction with SoMP is maintained (Fig. 5). The domains relocate with respect to each other but, except for the PKD2–PKD2 interface (Supplementary Fig. 6a, b), the dimer connections formed by individual domains do not change substantially. The β-propeller itself undergoes a conformational change, that includes the NGF binding site, in which opposing blades 5 and 10 have moved towards each other with rmsd of 1.6 Å over 499 Cα atoms (Supplementary Fig. 6c). This change seems to couple to a 9.9-degree rotation of the SoMP domain in the other molecule as to maintain the β-propeller – SoMP interaction and via SoMP to repositioning of the other domains in the dimer. The differences in the structures indicate that SorCS2 has substantial structural plasticity. Possibly, binding of NGF to SorCS2 triggers the

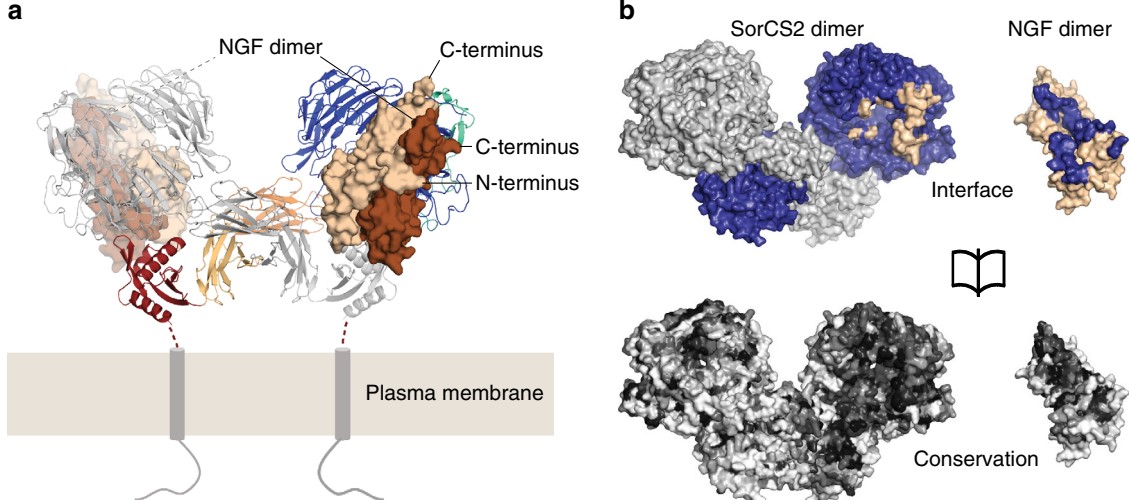

**Fig. 2** NGF binds to the top face of the sSorCS2 β-propeller to form a 4:2 complex. **a** The sSorCS2–NGF complex with the cell surface indicated. An NGF dimer is bound to each sSorCS2 chain. SSorCS2 is depicted in cartoon representation (colored according to Fig. 1b) and NGF in surface representation with the two dimer chains colored differently. **b** Opened view of the SorCS2–NGF interface (top, with one sSorCS2 monomer colored blue and the other gray) and the Consurf gradient conservation plot (based on SorCS members 1 to 3 and on proneurotrophins) on the surface of the sSorCS2–NGF complex (bottom), from white (not conserved) to black (highly conserved). The conservation of the interaction sites in both SorCS2 and NGF suggest the sSorCS2–NGF structure represents a common binding mode for other SorCS members and proneurotrophins

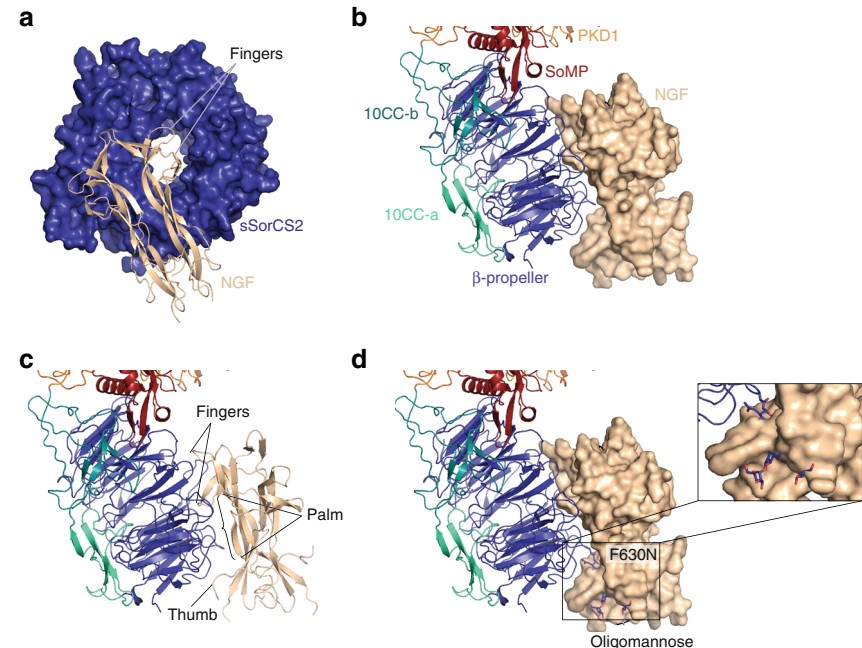

**Fig. 3** The NGF–sSorCS2 interaction resembles a hand holding a tire. **a** NGF (cartoon representation, beige) interacts with the top face of the sSorCS2 β-propeller (surface representation, blue) that resembles a tire. For clarity only the sSorCS2 β-propeller and NGF are shown. The two NGF fingers (formed by β-strands A′ and A″ and the loop connecting strands C and D) are positioned above the central β-propeller tunnel. **b**, **c** The NGF fingers, palm, and thumb are indicated. SSorCS2 in cartoon representation, colored according to Fig. 1a. NGF in surface representation in (**b**) and cartoon representation in (**c**). **d** Location of the F630N mutation and model of how the introduced oligomannose would interfere with NGF binding

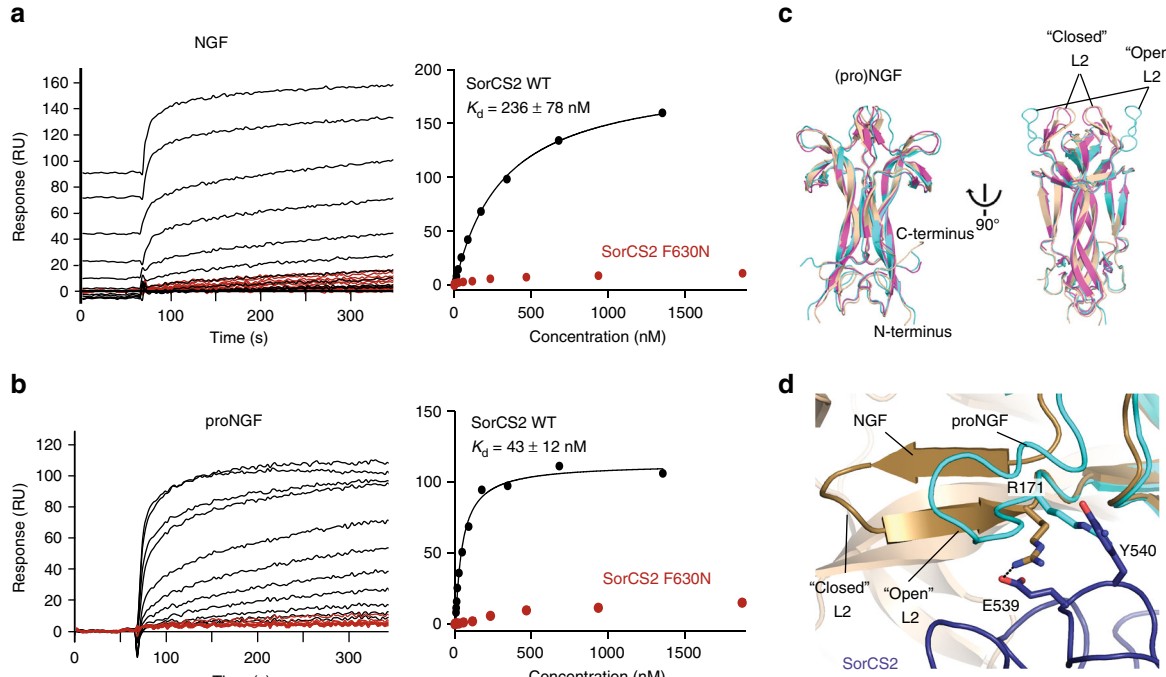

**Fig. 4** Both pro- and mature NGF directly bind SorCS2 on top of the β-propeller. **a**, **b** NGF (**a**) and proNGF (**b**) bind directly to sSorCS2 and interaction is abrogated by the F630N interface mutation on sSorCS2. Surface Plasmon Resonance sensorgrams of the association phase (left) and equilibrium binding plot (right) of sSorCS2 (black) and sSorCS2 F630N (red) binding to NGF and proNGF. Incomplete regeneration for NGF was accounted for by treating the data as an equilibrium titration experiment[62]. **c** SorCS2-bound NGF adopts the same conformation as in the 1:2 p75NTR–NGF complex[22]. Superposition of NGF from SorCS2–NGF (beige), the mature domain of proNGF from the symmetric p75NTR-proNGF structure (pdb-id 3IJ2[12], cyan) and NGF from the asymmetric p75NTR–NGF structure (pdb-id 1SG1[22], pink) in cartoon representation. In the SorCS2-NGF structure, NGF adopts a "closed" conformation[12] of the L2 loop, comparable to that in the asymmetric p75NTR–NGF structure. **d** Superposition of proNGF with an "open" conformation of the L2 loop (pdb-id 3IJ2, cyan) onto the SorCS2–NGF structure shows that the "open" conformation would clash between R171 and Y540, while in the "closed" conformation[12] R171 is stabilized by a charged interaction with E539

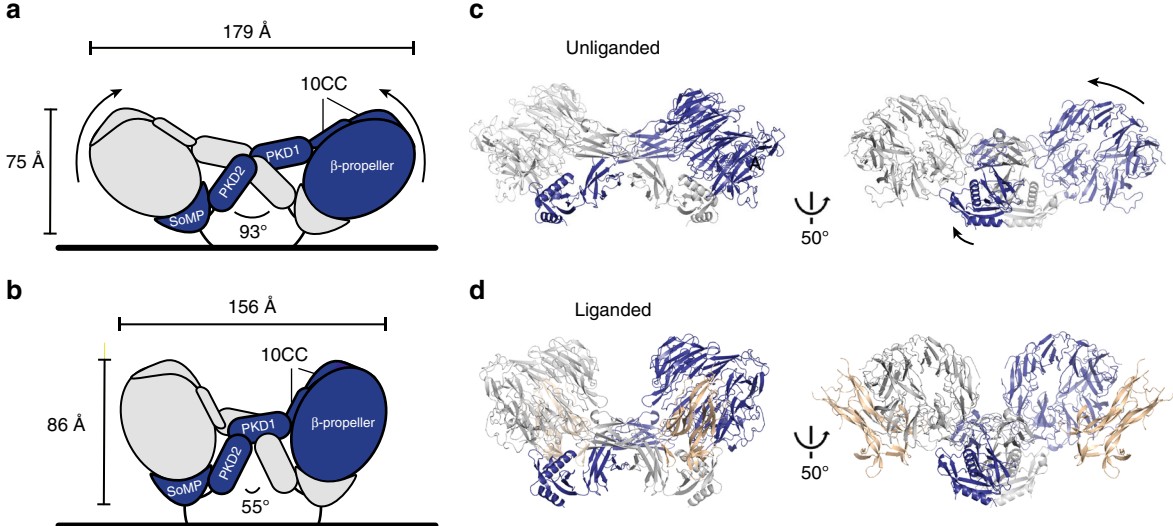

**Fig. 5** sSorCS2–NGF and unliganded sSorCS2 adopt different conformations. SorCS2 in complex with NGF has a different conformation and is more compact compared to the unliganded sSorCS2 structure. **a**, **b** domain rearrangement scheme of the conformational plasticity between unliganded (**a**) and liganded SorCS2 (**b**). Dimensions of the sSorCS2 dimers and angle between the PKD2 domains are indicated. NGF has been omitted for clarity. **c** cartoon representation of the unliganded SorCS2 homodimer in an orientation similar to (**a**) and in a 50° rotated view. The β-propellers are positioned at an angle with their ligand-binding top face close to the cell surface. **d** sSorCS2–NGF complex representation identical to (**c**). NGF is now shown (colored beige). In the NGF-bound structure, the β-propellers have moved up and away from the cell surface exposing the top face

conformational change within the β-propeller and indirectly the more substantial changes in the SorCS2 dimer. However, the similarity of our unliganded and liganded sSorCS2 crystal structures to two unliganded negative stain low-resolution sSorCS2 reconstructions determined recently[19] (Supplementary Fig. 7) suggests NGF may bind to SorCS2 by a conformational selection mechanism. In any case, the conformational plasticity observed is likely required to enable NGF binding to a membrane attached SorCS2 dimer as repositioning of the β-propellers in the complex away from the cell surface should alleviate steric hindrance of NGF with the membrane (Fig. 5 and Supplementary Fig. 8).

## Discussion
The cross-braced homodimer architecture and the domain composition of the VPS10p subunit followed by two PKD domains and a SoMP domain is predicted to be conserved among the three SorCS members. The sequence identity is at least 47% between the mature extracellular segments of SorCS2 and the other SorCS members (Supplementary Fig. 1) and the low-resolution negative stain electron microscopy reconstructions of sSorCS1, sSorCS2, and sSorCS3 dimers[19] resemble the crystal structures of the sSorCS2 dimers (Supplementary Fig. 7). These observations indicate that the cross-braced homodimer is a defining structural feature of the SorCS subfamily.

The SorCS2 and SorCS2-NGF structures reveal how the mature domain of proNGF interacts with SorCS2 but it does not inform on the contribution of the pro domain to binding. We show that the SorCS2 F630N mutation, in the NGF binding site, affects binding of NGF, proNGF, and proBDNF alike which indicates all three proteins binding sites should at least overlap. NGF and proNGF bind in a similar manner to p75NTR[12,22], and most likely binding to SorCS2 is similar for NGF and the mature domain of proNGF. The proNGF pro domain does, however, play an important role. It increases the affinity for SorCS2 (Fig. 4)[4] and Sortilin[6] and is generally believed to be required to trigger apoptotic signaling by the SorCS2/Sortilin–p75NTR complex[4,6]. In addition, a common single-nucleotide polymorphism in the pro domain of BDNF renders the isolated pro domain sufficient

to trigger growth cone retraction in hippocampal neurons that is dependent on p75NTR and SorCS2 expression[27]. The pro domain of proneurotrophins is intrinsically disordered which precludes us to predict where it would interact on SorCS2. It is, however, likely that the SorCS2 VPS10p subunit plays an important role, as the N-terminal residue of NGF is located close to the β-propeller and 10CC-a domain but away from other SorCS2 domains.

Several models of how proneurotrophins trigger apoptotic signaling have been suggested. P75NTR has been shown to be present in different oligomeric forms on the cell surface; as monomer, as covalent disulfide linked dimers, and as trimers or larger oligomers[24,30–32]. It has been suggested that p75NTR dimerizes[32,33], that p75NTR dimers undergo a conformational change[12,30] or that p75NTR monomerizes to trigger signaling[31]. In addition, it has been suggested that p75NTR trimers represent an inactivated form[31]. Importantly, the presence of SorCS2 (or Sortilin) is required for proNGF signaling and signaling is impaired if p75NTR is absent or if binding of proNGF to SorCS2 is blocked by an anti-SorCS2 antibody[3]. SorCS2 and p75NTR have been shown to interact independent of proNGF when expressed as full length transmembrane version[3,4] and this complex is associated with the intracellular guanine nucleotide exchange factor Trio that is released upon extracellular proNGF binding to SorCS2 and p75NTR to trigger neuronal growth cone retraction[3]. Finally, proNGF binds better to cells that express both receptors simultaneously compared to cells that express the receptors individually, indicating that a ternary complex consisting of SorCS2, p75NTR and proNGF is most likely formed[4]. How the structure of the SorCS2–NGF complex may shed new light on the signaling of proNGF via SorCS2 and p75NTR is outlined below.

In agreement with previous models[3], the structure of the sSorCS2–NGF complex indicates that proNGF signal induction is best explained by formation of a ternary SorCS2–proNGF–p75NTR signaling complex in which proNGF acts as a wedge in-between SorCS2 and p75NTR. The SorCS2 interaction site on proNGF overlaps with the known interaction sites of the p75NTR and TrkA co-receptors[12,22,23] (Fig. 6). In the sSorCS2-NGF complex one of the two equivalent receptor

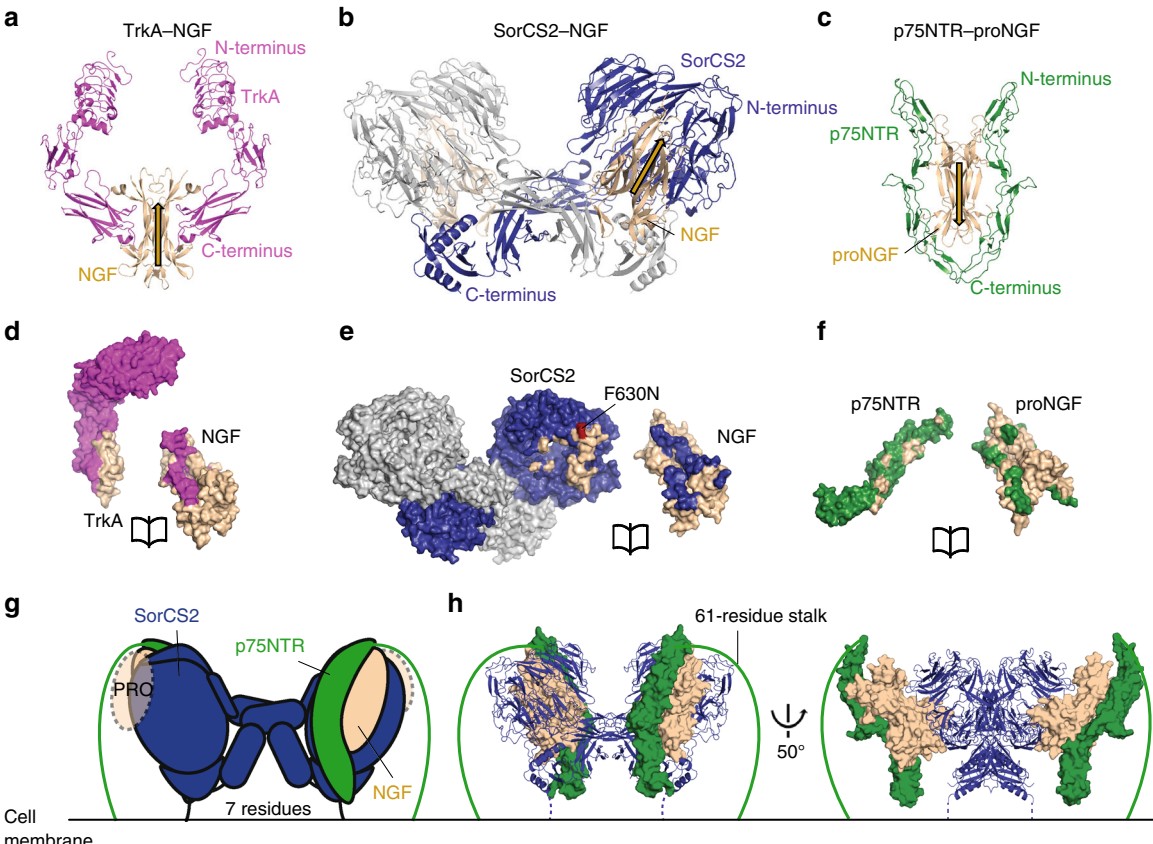

**Fig. 6** (pro)NGF complex structures reveal a common receptor binding site on NGF. **a–c** cartoon representation of NGF (beige) in complex with TrkA (**a**, pink) (pdb-id 2IFG)[24] and SorCS2 (**b**, blue-gray), and of proNGF (beige) in complex with p75NTR (**c**, green) (pdb-id 3IJ2)[12]. The receptors are depicted with their C-termini towards the bottom to indicate their cell-surface attachment side. The direction of (pro)NGF in the complexes is indicated with beige arrows to highlight the reversed direction of proNGF in complex with p75NTR. **d–f** open-book view of each receptor–ligand interface reveals that the receptor binding site for the three receptors, TrkA (**d**), SorCS2 (**e**), and p75NTR (**f**), all have overlap on the mature part of (pro)NGF. The interface mutant F630N (red) abrogates binding of NGF, proNGF, and proBDNF to SorCS2. **g**, **h** Possible model for a ternary SorCS2–proNGF–p75NTR complex. The proNGF pro domain is shown for clarity but its location and binding site in the complex are unknown. The pro domain does not interact with p75NTR[6]. The orientation of p75NTR is upside down in this model but the connection to the membrane may be bridged by the 61-residue long stalk. Domain representation scheme of the ternary model with SorCS2 in blue, the mature part of proNGF in beige, and p75NTR in green (**g**). Cartoon representation of the ternary complex model with coloring as in (**g**), the mature part of proNGF and the ligand binding segment of p75NTR (pdb-id 1SG1) in surface representation and the 61-residue p75NTR stalk represented by a line (**h**)

binding sites on NGF is free and exposed. Unless the conformational plasticity of SorCS2 is more substantial than apparent from the crystal structures, this second receptor binding site is unlikely to become occupied by a second SorCS2 dimer, as that would lead to steric clashes with the cell surface (Supplementary Fig. 9). A straightforward superposition of the sSorCS2–NGF and p75NTR–proNGF structures[12] (with one p75NTR molecule removed) based on the common mature NGF domain reveals that the free binding site can be occupied by p75NTR to form a ternary complex in *cis* on the same cell surface without steric clashes (Fig. 6g, h). We do not observe a ternary complex in solution (consisting of the SorCS2 and p75NTR extracellular regions and proNGF) nor do we observe direct SorCS2–p75NTR interactions in solution (Supplementary Fig. 10). Possibly, the absence of SorCS2–proNGF–p75NTR ternary complexes or direct SorCS2–p75NTR complexes may be explained by the lack of an avidity effect in solution that would be present on the cell surface if already interacting transmembrane proteins SorCS2 and p75NTR bind to proNGF simultaneously. This hypothesis is supported by the greatly enhanced binding of proNGF to cells that express both SorCS2 and p75NTR simultaneously compared to cells that express the receptors individually[4]. In addition, it has

been hypothesized that an asymmetric 1:2 p75NTR-(pro)NGF binding mode, as observed previously[22], is an intermediate state to enable ternary complex formation with the SorCS2 family member Sortilin[34].

In the SorCS2–proNGF–p75NTR co-receptor–ligand–receptor model the N-terminus of p75NTR is oriented towards the cell-surface similar to the upside-down orientation of p75NTR with respect to NGF-bound TrkA[23,24,35]. The extended 61-residue peptide stalk of p75NTR likely permits sufficient flexibility to bridge the connection to the cell surface. SorCS2 and p75NTR have been shown to be associated on the cell surface in the absence of proNGF[3,4] and it is believed that proNGF induces a separation between the co-receptors that triggers the release of the guanine nucleotide exchange factor Trio from the SorCS2–p75NTR complex cytosolic segments[3]. This hypothesis is supported by a ternary model in which the SorCS2 and p75NTR co-receptors are not interacting directly but are separated by the proNGF dimer that acts as a separating wedge. The importance of pre-association and ligand-induced conformational reorganization of receptors has been suggested for other signaling systems[36]. Possibly, the antiparallel orientation of p75NTR in the ternary SorCS2–proNGF–p75NTR complex may change the properties of

an inactive SorCS2–p75NTR complex or may restrict the location of the p75NTR peptide stalk and concomitantly its transmembrane helix and cytosolic domain away from the cytosolic segment of SorCS2 to induce signaling. However, further work will be required to resolve the details of how p75NTR, proNGF and SorCS2 interact in a ternary complex.

Members of the VPS10p family have been shown to play an important role in synaptic plasticity[4,5,37,38] and have been associated with neurodegenerative diseases such as Alzheimer's[39,40], frontotemporal lobar degeneration[41] and Huntington's[2], and psychological disorders such as schizophrenia, bipolar disorders, and attention-deficit hyperactivity disorder[42–44]. Since its structure was first solved, Sortilin has been the target for design of small-molecules inhibitors, in hopes that modulating its receptor function would offer therapeutic applications[45,46]. However, these efforts have been limited to targeting the only known binding site, present in the central tunnel of the β-propeller[14]. The structure of sSorCS2–NGF provides a blueprint for targeting a previously unobserved binding site on the top face of the β-propeller, although it is currently not experimentally verified whether proneurotrophins bind to the equivalent site on Sortilin. Nonetheless, the structures of sSorCS2 and the sSorCS2 complex presented here enable new options for structure-based inhibitor design, such as SorCS-subfamily specific inhibitors blocking a specific receptor conformation or preventing proneurotrophin binding. Future studies are required to determine how the pro domain of proneurotrophins can elicit such a drastic change in function, and what differences in signal transduction arise from the various proneurotrophins and VPS10p receptors complexes. The structures of unliganded sSorCS2 and the sSorCS2–NGF complex presented here provide a firm basis to start resolving these outstanding questions.

## Methods

**Generation of protein constructs and mutagenesis.** The construct of mouse SorCS2 (sSorCS2) was based on Image Clone 8861897 (Source Bioscience) and consisted of residues 117–1072 representing the mature extracellular segment. Constructs of proNGF (residues 19–241) and proBDNF (residues 19–249) were based on codon-optimized synthetic DNA (DNA 2.0) with all furin sites modified from RR/KR to AA (Supplementary Table 1). Mouse p75NTR was based on Image Clone 5367638 (Source Bioscience) and consisted of residues 32–252 representing the full extracellular segment (p75NTRfe). A sSorCS2 mutant (F630N) that introduces an N-linked glycosylation site in the proNGF binding site was generated for interaction experiments. DNA amplification was performed using primers from Supplementary Table 2. All constructs were subcloned using BamHI/NotI sites in the pUPE107.03 vector (U-Protein Express) containing a cystatin secretion signal peptide and a C-terminal His$_6$-tag. Mouse mature NGF was purchased from Bio-Rad.

**Protein expression and purification.** Recombinant proteins were produced either in Epstein–Barr virus nuclear antigen I (EBNA1)-expressing HEK293 cells (HEK293-E) or in N-acetylglucoaminyltransferase I-deficient (GnTI−) EBNA1-expressing HEK293 cells (HEK293-ES) (U-Protein Express). Proteins produced in HEK293-ES cells contain shorter, more homogeneous oligo-mannose glycans. Proteins produced in HEK293-E cells contain hybrid glycans. For crystallization and SAXS experiments proteins were produced in HEK293-ES cells. For SPR, all proneurotrophins and sSorCS2 in Fig. 4a, b was from HEK293-ES, sSorCS2 in Supplementary Fig. 5 was from HEK293-E. For MALS, proNGF and sSorCS2 were from HEK293-ES and p75NTRfe was from HEK293-E (Supplementary Fig. 10 and Supplementary Fig. 3a), sSorCS2 in Supplementary Fig. 3b was from HEK293-E. Medium was harvested 6 days after transfection and cells were spun down by 10 min of centrifugation at 1000 × g. Supernatant was concentrated fivefold and diafiltrated against 500 mM NaCl, 25 mM HEPES pH 7.5 (IMAC A) using a Quixstand benchtop system (GE Healthcare) with a 10 kDa molecular weight cut-off (MWCO) membrane. Cellular debris were spun down for 10 min at 9500 × g and the concentrate was filtered with a glass fiber prefilter (Minisart, Sartorius). Protein was purified by Nickel-nitrilotriacetic acid (Ni-NTA) affinity chromatography using IMAC A for binding to the Ni-NTA column and step-eluted with 4, 6 and 40% IMAC B buffer containing 25 mM HEPES pH 7.5, 500 mM NaCl, 500 mM imidazole. Affinity chromatography was followed by size exclusion chromatography (SEC) on a Superose 6 Hiload 16/60 column (GE Healthcare) equilibrated in SEC buffer (for proneurotrophins: 20 mM HEPES pH 7.0, 150 mM NaCl; for

sSorCS2: 25 mM 2-(N-morpholino)ethanesulfonic acid (MES) pH 5.5, 500 mM NaCl). Wt sSorCS2 was concentrated to 12.6 mg mL$^{-1}$, mutant sSorCS2 F630N was concentrated to 19.2 mg mL$^{-1}$, using a 30 kDa MWCO concentrator before plunge freezing in liquid nitrogen and storage at −80 °C. ProNGF was concentrated to 11.3 mg mL$^{-1}$ and proBDNF to 9.9 mg mL$^{-1}$, using a 10 kDa MWCO concentrator before plunge freezing in liquid nitrogen and storage at −80 °C.

**Crystallization of mouse sSorCS2 and the sSorCS2–NGF complex.** For the sSorCS2 crystals, sSorCS2 was deglycosylated using EndoHf 1:100 O/N at RT in buffer pH, and limited proteolysis with trypsin 1:100 was performed for 20 min before setting up crystallization. For crystallization of the sSorCS2–NGF complex, sSorCS2 and proNGF samples were mixed and diluted to a sSorCS2 concentration of 80 µM with 20 mM HEPES pH 7.0, 150 mM NaCl, with a ratio of 1:1.1 sSorCS2: proNGF, and Calcium chloride was added to a final concentration of 1 mM. Sitting-drop vapor diffusion at 18 °C was used for all crystallization trials, by mixing 150 nL of protein solution with 150 nL of reservoir solution. Crystal appeared after three weeks but the proNGF in the crystallization drop revealed degradation to NGF. No electron density is apparent for the pro domain of proNGF in the diffraction data and we therefore assume the complex crystallized is sSorCS2–NGF (Supplementary Fig. 11). Crystallization trials with mature NGF and sSorCS2 did not yield any crystals. SSorCS2 crystals were obtained from a condition containing 0.1 M Sodium chloride, 0.02 M Tris pH 7.5, 0.1 M Magnesium chloride hexahydrate, and 11 % w/v PEG 1500, final pH 6.4. Complex crystals were obtained from a condition containing 0.225 M MES/bis-tris pH 6.6 and 6.6 % w/v PEG 6000. Crystals were harvested and flash-cooled in liquid nitrogen in the presence of reservoir solution supplemented with 25 % ethylene glycol.

**X-ray data collection.** Diffraction data were collected up to 4.2 Å resolution for the sSorCS2 dataset and 3.9 Å resolution for the sSorCS2–proNGF dataset at 100 K at European Synchrotron Radiation Facility (ESRF) beamlines ID29 and ID23-2, respectively.

**Structure determination and refinement.** Data was integrated using XDS[47] and Mosflm[48], respectively and further processed in AIMLESS[49]. Resolution limits were determined by applying a cut-off based on the mean intensity correlation coefficient of half-datasets, CC$_{1/2}$. Molecular replacement on the sSorCS2–proNGF dataset was performed in Phaser[50] using search models with PDB codes 4EAX (mouse NGF)[51], 4MSL (human Sortilin)[52], 1WGO (PKD domain), and 4AQO (PKD domain)[53]. Sculptor[54] was used to improve the molecular replacement models. Initial refinement was performed in Refmac[55]. Separate domains from this sSorCS2 partial model were used as templates for molecular replacement using Phaser in the sSorCS2 dataset. Density modification with multi-crystal averaging, using four rigid-bodies over three independent molecules of the sSorCS2–proNGF and sSorCS2 datasets was performed in Phenix[56]. This improved the density of the unknown C-terminal domain and the rest of the sSorCS2 moiety in the data of the sSorCS2–proNGF complex sufficiently to allow manual model (re)building in Coot[57]. Further refinement was performed in Refmac using Prosmart external restraints[58], followed by several cycles of Phenix-Rosetta[59] carried out on the Surfsara life science cluster, and alternate model building in Coot and refinement in Refmac and Phenix with enforcement of secondary structure restraints. Final refinement of the sSorCS2–proNGF complex was performed in Phenix. Individual domains of the refined sSorCS2 structure of the sSorCS2–proNGF complex were placed in the unliganded sSorCS2 data and refined using several hundred cycles of jelly body restraints in Refmac. This was followed by limited manual rebuilding in Coot and further refinement in Phenix and Refmac. Final refinement of the unliganded sSorCS2 structure was performed in Phenix. Molprobity[60] was used for structure validation. Figures were generated with PyMol (Schrödinger) or UCSF Chimera[61].

**Surface plasmon resonance.** Equilibrium binding studies were performed using an MX96 instrument (IBIS Technologies). NGF, ProNGF and proBDNF at 200 µg mL$^{-1}$ were amine-coupled for 45 min at pH 4.5 to a planar-type P-COOH SensEye SPR sensor (IBIS Technologies) after 1-ethyl-3-(3-dimethylaminopropyl) carbodiimide hydrochloride/N-Hydroxysuccinimide (EDC/NHS) activation. In addition, NGF was also coupled at pH 5, 5.5, and 6 but no observable differences in coupling or affinity were found. Wt and mutant sSorCS2 were flowed over the sensor chip, as analyte, in buffer containing 25 mM HEPES pH 7.4, 150 mM NaCl, and 0.005 % Tween 20. Temperature was kept constant at 25 °C. NaI at 1.0 M was used as regeneration condition. The data was analyzed using SprintX (IBIS Technologies) and Prism. A one site specific saturation model was used on the association phase averaged data (between 320 and 340 s) to determine the dissociation constant ($K_d$) and the maximum analyte binding ($B_{max}$). Incomplete regeneration of NGF was accounted for by zeroing the data only at the start of the injection series[62]. SigmaPlot two site saturation model was used for proBDNF data on the association phase averaged data (between 320 and 340 s) to determine the dissociation constant ($K_d$) and the maximum analyte binding ($B_{max}$).

**Size exclusion chromatography multi-angle light scattering.** SEC multi-angle light scattering (SEC-MALS) was used to determine the oligomeric state of

sSorCS2. Ten microlitres of 10 mg mL$^{-1}$ sSorCS2 was injected onto a Superose 6 5/150 gel filtration column (GE Healthcare) in 25 mM MES pH 5.5, 150 mM NaCl or a Superose 6 10/300 gel filtration column (GE Healthcare) in 25 mM HEPES pH 7.4, 150 mM NaCl, and 2 mM CaCl for all other experiments. For molecular weight characterization, light scattering was measured with a miniDAWN TREOS multi-angle light scattering detector (Wyatt), connected to a differential refractive index monitor (Shimadzu, RID-10A) for quantitation of the protein concentration. Chromatograms were collected, analyzed, and processed by ASTRA6 software (Wyatt, using calculated dn/dc values of 0.183 mL g$^{-1}$ for sSorCS2 (taking 6w/w % glycosylation into account), 0.184 mL g$^{-1}$ for proNGF, and 0.173 mL g$^{-1}$ for p75NTR). The calibration of the instrument was verified by injection of 10 µL of 10 mg mL$^{-1}$ monomeric bovine serum albumine (BSA, Sigma-Aldrich) or 100 µL of 1 mg mL$^{-1}$ conalbumin.

**SEC-SAXS measurements and data collection**. SAXS experiments were performed at ESRF beamline BM29 in Grenoble. A Superose 6 10/30 column (GE Healthcare) connected to a high-performance liquid chromatography (HPLC) system was equilibrated in 1.5 CV of 25 mM MES pH 5.5 and 150 mM NaCl. A volume of 40 µL wild type sSorCS2 at 12.6 mg mL$^{-1}$ was loaded on the column. The HPLC system consists of an in-line degasser (DGU-20A5R, Shimadzu, France), a binary pump (LC-20ADXR, Shimadzu, France), a UV-VIS array photospectrometer (SPD-M20A, Shimadzu, France) and a conductimeter (CDD-10AVP, Shimadzu, France) attached to the sample-inlet valve of the BM29 sample changer. The measurements were performed at room temperature with a flow rate of 0.6 mL min$^{-1}$. 2400 frames (1 frame s$^{-1}$) were collected at a wavelength of 0.9919 Å using a sample-to-detector (PILATUS 1 M, DECTRIS) distance of 2.81 m. The EDNA pipeline was used to perform automatic initial data processing. After selecting 10 frames with a consistent $R_g$ from the peak scattering intensity, GNOM (EMBL-HH ATSAS suite)[63] was used to automatically merge these frames to yield an averaged frame that corresponds to the scattering of sSorCS2. Data was further analyzed by PRIMUS[64] and GNOM of the ATSAS suite[63]. The error estimate for the $R_g$ value calculated by Guinier analysis is the standard error for linear regression. The error for the maximum particle dimension, $D_{max}$, is representative for the optimal-solution range. Experimental and calculated scattering curves from sSorCS2 crystal structures were fitted using Crysol[65].

**Data availability**. Atomic coordinates and structure factors have been deposited in the Protein Data Bank (PDB) with accession number 6FG9 [https://doi.org/10.2210/pdb6FG9/pdb] and 6FFY [https://doi.org/10.2210/pdb6FFY/pdb] for sSorCS2 and the sSorCS2–proNGF complex, respectively. All other data are available from the corresponding author upon reasonable request.

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

## Acknowledgements
We thank the staff of ESRF beamlines ID29, ID23-2, and BM29. This work was funded by the Initial Training Network grant "ManiFold" from the EU under FP7 (grant agreement number 317371) and an Investment Grant (721.012.004) from the Netherland Organization for Scientific Research (NWO). B.J.C.J. is supported by an NWO Vidi grant (723.012.002) and by a European Research Council Starting Grant (677500).

## Author contributions
N.L. and B.J.C.J. designed the experiments. L.M.P.C. cloned and purified the full ecto-domain of p75NTR and sSorCS2 F630N mutant, performed the SPR experiments with NGF and proNGF, MALS experiments at pH 7.4, and crystallization drop analysis. N.L. performed all other experiments. B.J.C.J. solved and refined the crystal structures and supervised the project. N.L and B.J.C.J analyzed the structural information and wrote the manuscript to which L.M.P.C. commented.

## Additional information

**Competing interests:** The authors declare no competing interests.

