## [Peer Review File · Nature Communications]

Reviewers' comments:

Reviewer #1 (Remarks to the Author):

This manuscript by Lecoup and colleagues evaluates the structure of the sortilin family member, SorCS2, a receptor best characterized for its roles in regulating neuronal morphology and synaptic plasticity following engagement with proneurotrophins. This receptor family has recently been the focus of significant investigation, given its potential roles in neuropsychiatric conditions. Here the authors describe the crystal structure of SorCS2 alone, and in complex with proNGF. While these structures provide a better understanding of the structure and dimeric nature of SorCS2, significant questions remain as to the role of ligand engagement in altering SorCS2 receptor conformation and in initiating signaling, known to require both SorCS2 and the unrelated receptor component, p75. These studies build on very recent publications which document that SorCS receptors, including SorCS2, are present in two different dimeric conformations (more compact, and more extended) in the absence of ligand. Additional biochemical analyses are required to understand how the conformational flexibility of the SorCS family of receptors regulates function.

1. The current study describes two crystal structures of SorCS2, one in the absence of ligand and one in complex with proNGF. As recent work indicates that SorCS2 exists in two different dimeric states (compacted or elongated, in the absence of ligand, Januliene et al, 2017), it is important to provide evidence as to whether proNGF induces these changes, or simply "selects" a SorCS2 dimer state that appropriately packs into a crystal lattice. As the sites of interaction of the mature NGF domain, visualized in the crystal structure, are far removed from the domains of SorCS2 which undergo significant conformational shifts (SoMP:beta propeller, PKD1:PKD2), it is unclear whether the mature NGF domain interactions with SorCS2 are driving this, or whether undetected prodomain: SorCS2 interactions are crucial. The authors describe large conformational changes of 30 degree angle rotation, and detail contact areas between the dimers of more than 5000 square angstroms. Is a SorCS2 monomer so rigid that engagement of the mature domain of proNGF in the beta propeller is able to transmit a conformational change in multiple distinct domains? Is mature NGF alone able to do this, or is the prodomain required? Although the prodomains of neurotrophins are disordered, prior reports define the specific regions of the BDNF prodomain which interact with SorCS2 and sortilin, suggesting that the prodomain may contribute to the interaction. The authors, in the title and in the paper, imply that proNGF induces a conformational change to initiate signaling. These mechanistic questions need to be resolved.

2. The region of proNGF which is resolved in the current crystal structure resembles mature NGF (with a "closed" L2 loop structure), not the crystal structure of proNGF in complex with p75 (Feng et al, 2010), with "open" L2 loops which could accommodate some regions of the prodomain. This raises the question of whether proNGF is intact in the crystal, and blots should be provided. This is particularly important as prior work by others suggest that the related SorCS3 binds to both mature NGF and proNGF (Westergaard et al, 2005), although there are no reports that mature neurotrophins induce signaling by SorCS receptors. To determine whether SorCS2 binds to mature NGF, proNGF or the prodomain, the authors should undertake Biacore analysis using immobilized SorCS2 ectodomain, to evaluate the relative affinities of mature NGF, proNGF or the isolated prodomain of NGF with SorCS2.

3. Prior studies indicate that effective proneurotrophin signaling requires co-expression of a sortilin/SorCS2 family member and p75, and the crystal structure of dimeric proNGF appears to be in complex with a dimer of p75 (Feng et al, 2010). The structures described here provide some significant challenges to the prior studies, and should be resolved, as (1) the binding surface of the mature domain of proNGF to SorCS2 would exclude p75 dimerization and (2) the orientation of proNGF is almost 180 degrees "off" the predicted proNGF-p75 orientation in which the L2 loops are likely to face away from the membrane, not towards the membrane. As such, figure 3D is quite speculative, and supplementary fig 8 top row is likely to be misleading to a general audience, with the p75/proNGF model "upside down" as compared to the orientation of Trk or SorCS2 receptors. To clarify this, the authors should provide biochemical data to determine the stoichiometry of

SorCS2:proNGF:p75 in solution. This could be readily accomplished using gel filtration with soluble ectodomains of SorCS2 and p75, and proNGF. This approach would determine whether proNGF concomitantly engages SorCS2 and p75, and would provide useful information as to whether these interactions are dependent on the local environment (such as cations), as has been suggested for sortilin/p75/proNGF interactions. This data would significantly refine the field's understanding of how proneurotrophins signal (using SorCS2 dimers and "passing" to p75, or engaging SorCS2 dimers and a p75 monomer, which would support the preliminary model proposed).

4. The question of SorCS2 dimerization is approached only at low pH (5.5), using chromatography and light scattering. This should be repeated at neutral pH, as SorCS2 is cell surface exposed and interacts with proneurotrophin ligands at neutral pH.

Reviewer #2 (Remarks to the Author):

NCOMMS-17-33077-T

Review comment:

Leloup et al., "Structure of the SorCS2-proNGF signaling complex reveals a conformation-dependent mechanism regulating neuronal plasticity"

Sortilin and its related proteins are type-1 membrane receptors implicated in neuronal viability and function. Their extracellular domains are characterized by a ten-bladed beta-propeller known as VPS10p domain. Sortilin family proteins are endocytic receptors, which traffic through the trans-Golgi network, endosomal compartments, lysosome, and cell surface. Sortilin family proteins bind to pro-forms of neurotrophic factors (NTs) such as NGF and BDNF as ligands. In addition, the co-receptor p75NTR functions with the Sortilin family proteins to transmit downstream signals into the cytoplasm. Owing to the lack of their complex structures with NTs, their underlying structural mechanisms remain elusive. In this study, Leloup et al. present the crystal structures of the extracellular domain (ECD) of Sortilin-related CNS-expressed receptor 2 (SorCS2) alone and its complex proNGF at 4.0 and 3.6 Å resolutions, respectively. The SorCS2 ECD consists of the VPS10p, two Cys-rich 10CC, and two polycystic kidney disease (PKD), Fibronectin type-III (FN-III), and unpredicted membrane proximal (SoMP) domains. Both the unliganded and liganded SorCS2 structures exhibit a cross-braced dimer but in different conformations, illustrating the conformational change in the SorCS2 ECD upon binding to proNGF, which seems to be required for binding to the co-receptor p75NTR. The dimeric SorCS2 binds to two proNGF dimers via the VPS10p domain. The vacant side of proNGF can be utilized for the binding to p75NTR. The two PKD domains mediate the dimer interface, which slides upon binding to proNGF. The structures provide new structural insights into the recognition of NTs by SorCS2 and important clues to understand the structural mechanism for the activation of downstream events including the binding to p75NTR. However, it seems that descriptions on physiological roles and molecular mechanisms of the Sortilin family are insufficient for general readers to understand the importance or interesting points of this study. I would like to suggest the following points to be improved for publication:

1. Introduction, Results, and Discussion sections should be separated. Results section should also be divided into subsections with subheadings to clarify what the authors present.

2. The current introduction part lacks the details of the functional roles of the Sortilin family proteins in the context of molecular signaling for the regulation of cellular events. How do the Sortilin family proteins regulate synaptic plasticity and cell death/survival? How are neurotrophic ligands, p75NTR, and Trk involved in these events? What is the difference of SorCS1–SorCS3, Sortilin, and SorLA? Why do the Sortilin family proteins bind to proform NTs instead of mature NTs? In addition, the intracellular trafficking of the Sortilin family proteins and its physiological significance should be described. Citation of review article(s) (e.g., Nykjaer and Willnow, Trends in

Neuroscience, 35, 261-270 (2012)) may be helpful.

3. Descriptions of the unliganded and liganded structures should be separated, because these two structures exhibit different conformations. I would like to suggest the authors to make a subsection for each structure.

4. pg 4, How did the authors define the top and bottom of the beta-propeller? Should be explained.

5. The liganded structure suggests that the pro region of NGF or BDNF is dispensable for the binding to SorCS2. Comparison between the affinities to the pro-form and mature form of NTs should be required to confirm that the pro region is not actually involved in the binding.

6. pg 5, "Biophysical interaction assays ..." may be rewritten as "Surface plasmon resonance analyses ..." to specify the method.

7. pg 5, The structure around Phe630 of SorCS2 should be presented to show how the introduced N-linked glycan affects the interface.

8. SEC-MALS and/or SAXS analyses of the SorCS2–proNGF–p75NTR ternary complex may support the proposed structural mechanism of the ternary complex assembly.

9. The difference or similarity between SorCS2 and other Sortilin family proteins should be discussed at primary or tertiary structure levels. In addition, discussion regarding future studies and/or applications utilizing the present structures may be helpful for general researchers in biological or medical science fields.

10. When resolution limits are defined on the basis of $CC(1/2)$, $CC(1/2)$ in the high resolution shell is typically more than 0.5. I would like to suggest the authors to try structure refinement at lower resolution with $CC(1/2)$ in the high resolution shell being more than 0.5. If such refinement results in worse geometry, the current refinement at 3.6 (liganded) or 4.0 Å (unliganded) resolution may be OK. If not, the effective resolution may be worse than 3.6 or 4.0 Å resolution, because the diffraction data corresponding to high resolution seem to have little structural information.

11. pg 17., PDB IDs are missing.

12. Figure legends of Supplementary Fig.3, "... indicates a molecular mass of ..." should be "... indicates a molar mass of ...".

13. There are several typos (e.g., "... in the in the crystal lattice for ..." in figure legends of Supplementary Fig. 4). Check the manuscript more carefully before resubmission.

14. The PDB validation reports are inconsistent with Table 1. The outlier residues in Ramachandran plot should be corrected unless their unfavorable main-chain conformations are stabilized by some interactions.

We would like to thank the reviewers for their excellent comments. Based on their review we have performed additional experiments and have revised the text and figures to implement the additional insights and to enhance the clarity and scope of the manuscript. Below we have outlined our response to the reviewers' comments and detailed our changes in the manuscript.

Reviewers' comments:

Reviewer #1 (Remarks to the Author):

This manuscript by Leloup and colleagues evaluates the structure of the sortilin family member, SorCS2, a receptor best characterized for its roles in regulating neuronal morphology and synaptic plasticity following engagement with proneurotrophins. This receptor family has recently been the focus of significant investigation, given its potential roles in neuropsychiatric conditions. Here the authors describe the crystal structure of SorCS2 alone, and in complex with proNGF. While these structures provide a better understanding of the structure and dimeric nature of SorCS2, significant questions remain as to the role of ligand engagement in altering SorCS2 receptor conformation and in initiating signaling, known to require both SorCS2 and the unrelated receptor component, p75. These studies build on very recent publications which document that SorCS receptors, including SorCS2, are present in two different dimeric conformations (more compact, and more extended) in the absence of ligand. Additional biochemical analyses are required to understand how the conformational flexibility of the SorCS family of receptors regulates function.

1. The current study describes two crystal structures of SorCS2, one in the absence of ligand and one in complex with proNGF. As recent work indicates that SorCS2 exists in two different dimeric states (compacted or elongated, in the absence of ligand, Januliene et al, 2017), it is important to provide evidence as to whether proNGF induces these changes, or simply "selects" a SorCS2 dimer state that appropriately packs into a crystal lattice. As the sites of interaction of the mature NGF domain, visualized in the crystal structure, are far removed from the domains of SorCS2 which undergo significant conformational shifts (SoMP:beta propeller, PKD1:PKD2), it is unclear whether the mature NGF domain interactions with SorCS2 are driving this, or whether undetected prodomain: SorCS2 interactions are crucial. The authors describe large conformational changes of 30 degree angle rotation, and detail contact areas between the dimers of more than 5000 square angstroms. Is a SorCS2 monomer so rigid that engagement of the mature domain of proNGF in the beta propeller is able to transmit a conformational change in multiple distinct domains? Is mature NGF alone able to do this, or is the prodomain required? Although the prodomains of neurotrophins are disordered, prior reports define the specific regions of the BDNF prodomain which interact with SorCS2 and sortilin, suggesting that the prodomain may contribute to the interaction. The authors, in the title and in the paper, imply that proNGF induces a conformational change to initiate signaling. These mechanistic questions need to be resolved.

A: We have reanalyzed the two low-resolution negative stain electron microscopy reconstructions of unliganded SorCS2 (Januliene et al., 2017) and compared the reconstructions with maps based on our crystal structures filtered to 20 Å resolution. The crystal structure of unliganded SorCS2 fits well with EMD-id 3710 (of unliganded SorCS2), although the EM map is slightly more extended. The crystal structure of ligand bound SorCS2 (with the ligand removed) fits reasonably well with EMD-id 3840 (of unliganded SorCS2) although the β-propellers are rotated differently and the PKD1-PKD2-

SoMP dimer subunit is less resolved in the EM map. The differences between the two EM maps are larger than the differences between the two crystal structures which indicate that SorCS2 has intrinsic structural plasticity, also in the absence of ligand, and can perhaps adopt a range of conformations. The differences in conformation we observe in the crystal structures most probably represent two of these SorCS2 states. In addition the two SorCS2 dimer chains in the unliganded structure are also different to each other (Suppl. Fig. 2). Clearly, SorCS2 is not a rigid molecule and the crystal structures provide the first detailed insights into the conformations and the plasticity inherent to this receptor. Possibly, NGF does not induce a conformational change in SorCS2 but may select a SorCS2 state, although it seems unlikely that NGF would bind to the SorCS2 conformation (or a similar conformation) that we observe in the unliganded crystal structure as that would likely lead to steric hindrance with the cell surface (Suppl. Fig. 8). We have adjusted the manuscript, in particular the title, abstract and discussion, to emphasize that SorCS2 displays structural plasticity and discuss the conformational selection and conformational change models in the manuscript.

2. The region of proNGF which is resolved in the current crystal structure resembles mature NGF (with a “closed” L2 loop structure), not the crystal structure of proNGF in complex with p75 (Feng et al, 2010), with “open” L2 loops which could accommodate some regions of the prodomain. This raises the question of whether proNGF is intact in the crystal, and blots should be provided. This is particularly important as prior work by others suggest that the related SorCS3 binds to both mature NGF and proNGF (Westergaard et al, 2005), although there are no reports that mature neurotrophins induce signaling by SorCS receptors. To determine whether SorCS2 binds to mature NGF, proNGF or the prodomain, the authors should undertake Biacore analysis using immobilized SorCS2 ectodomain, to evaluate the relative affinities of mature NGF, proNGF or the isolated prodomain of NGF with SorCS2.

A: We thank the reviewer for raising this important point. Unfortunately it has been difficult to reproduce the complex crystals and we were unable to test the composition of the crystal on a SDS PAGE gel. Instead, we analysed the components of an identical crystallization setup incubated for more than three weeks (which was the time that the crystals needed to grow) on SDS PAGE gel stained with Coomassie Brilliant Blue. This shows that, despite mutating all the proNGF furin sites (RR or KR) to AA and not having noticed degradation previously, most of the proNGF in the crystallization condition is processed into mature NGF. Most likely the NGF form that was captured in the crystals is mature NGF and not proNGF, even though we could not obtain crystals from sSorCS2 set up with NGF directly. We sincerely apologize for this oversight from our side and again thank the referee for asking to test this. This of course explains why the pro domain of proNGF was not resolved in our crystals and why NGF bound to SorCS2 has a “closed” L2 loop instead of an “open” loop as observed in proNGF bound to p75NTR (Feng et al., 2010). To verify that both NGF and proNGF bind to the same site on SorCS2 that we observe in our SorCS2-NGF crystal structure we tested mature NGF and proNGF (that was intact as assessed on SDS PAGE gel) for binding to SorCS2 and the SorCS2 interface mutant F630N (see also updated Figure 4A). This shows both NGF and proNGF interact with SorCS2 and that proNGF interacts with six-fold higher affinity to SorCS2 (Kd of 43 nM) than does mature NGF (Kd of 236 nM). The interaction is abrogated for both NGF and proNGF by the SorCS2 F630N mutation and indicates that the interaction site we observe in the SorCS2-NGF structure is also used by proNGF.

We have added the gel analysis of the crystallization condition to supplementary figure 11, The results of the SPR analysis to Figure 4 and have adjusted the manuscript to point out that the complex structure is SorCS2-NGF instead of SorCS2-proNGF. We believe previous analysis and

conclusions have not changed with the realization that mature NGF instead of proNGF crystallized in complex to SorCS2. We show that both NGF and proNGF interact at the same site on SorCS2 and that the pro domain contributes to the interaction with SorCS2 as was shown by others previously (Glerup et al., 2014). We have extended the discussion to emphasize that our data does not reveal how the pro domain of NGF contributes to interaction with SorCS2.

3. Prior studies indicate that effective proneurotrophin signaling requires co-expression of a sortilin/SorCS2 family member and p75, and the crystal structure of dimeric proNGF appears to be in complex with a dimer of p75 (Feng et al, 2010). The structures described here provide some significant challenges to the prior studies, and should be resolved, as (1) the binding surface of the mature domain of proNGF to SorCS2 would exclude p75 dimerization and (2) the orientation of proNGF is almost 180 degrees “off” the predicted proNGF-p75 orientation in which the L2 loops are likely to face away from the membrane, not towards the membrane. As such, figure 3D is quite speculative, and supplementary fig 8 top row is likely to be misleading to a general audience, with the p75/proNGF model “upside down” as compared to the orientation of Trk or SorCS2 receptors. To clarify this, the authors should provide biochemical data to determine the stoichiometry of SorCS2:proNGF:p75 in solution. This could be readily accomplished using gel filtration with soluble ectodomains of SorCS2 and p75, and proNGF. This approach would determine whether proNGF concomitantly engages SorCS2 and p75, and would provide useful information as to whether these interactions are dependent on the local environment (such as cations), as has been suggested for sortilin/p75/proNGF interactions. This data would significantly refine the field’s understanding of how proneurotrophins signal (using SorCS2 dimers and “passing” to p75, or engaging SorCS2 dimers and a p75 monomer, which would support the preliminary model proposed).

A: We have now performed size exclusion chromatography of SorCS2:proNGF:p75NTR ectodomains in solution as the reviewer suggests, and simultaneously analysed the runs with MALS. We used neutral pH and a buffer supplemented with Ca^{2+} as this was shown to be important for ternary complex formation with Sortilin (Feng et al. 2010). Our experiment shows that proNGF and p75NTR interact. SorCS2 however does not interact with either proNGF or p75NTR in this setting as there is no peak shift apparent for SorCS2, nor is there a mass increase. We observe a weight average mass of individual SorCS2 dimers and proNGF:p75NTR complexes that are co-eluting. The proNGF:p75NTR complexes probably exists as a mix of 2:1 and 2:2 complexes similar as observed previously (Feng et al., 2010). The absence of SorCS2:proNGF:p75NTR complexes may be explained by the lack of an avidity effect in solution that would be present on the cell surface if transmembrane proteins SorCS2 and p75NTR bind to proNGF simultaneously. This hypothesis is supported by cell-binding assays that show SorCS2 and p75NTR cooperate in proNGF binding. ProNGF interacts with higher affinity to cells expressing both SorCS2 and p75NTR on the surface compared to cells expressing SorCS2 and p75NTR individually (Glerup et al., 2014). Our data do not exclude a model in which cell-surface expressed SorCS2 binds proNGF and then passes it on to p75NTR to trigger signaling, however such a model is not in line with the increased affinity of proNGF for a SorCS2-p75NTR combination (Glerup et al., 2014).

SorCS2 and p75NTR have been shown to interact independent of proNGF when expressed as full length transmembrane version (Deinhardt et al., 2011 and Glerup et al., 2014) and p75NTR itself is present in different oligomeric forms on cells; as monomer, as covalent disulfide linked dimers or as trimers (Viral et al., 2009, Anastasia et al., 2015 and Nadezhdin et al., 2016). The molecular mechanism that triggers proNGF signaling is not resolved. It has been suggested that proNGF-induced p75NTR dimerization activates signaling (Nadezhdin et al., 2016 and Vilar et al., 2009) but also that it is proNGF-induced p75NTR monomerization that activates signaling and that p75NTR

trimers represent an inactivated form (Anastasia et al., 2015). The authors of the 2:2 proNGF:p75NTR crystal structure (Feng et al., 2010) also discussed that it is not clear if their symmetric structure represents the true cell-surface composition of an active p75NTR and that a conformational change in p75NTR may be required to induce signaling. The presence of SorCS2 (or Sortilin) is required for proNGF signaling and signaling is impaired if p75NTR is absent or if binding of proNGF to SorCS2 is blocked by an anti-SorCS2 antibody (Deinhardt et al., 2011). Our data does indicate that the SorCS2 interaction site on NGF overlaps partially with the binding site for p75NTR (and Trk). Therefore, proNGF participating in a ternary SorCS2:proNGF:p75NTR complex will not be able to bind a second p75NTR as in a symmetric 2:2 proNGF:p75NTR complex. In addition, the upside down orientation of p75NTR in the SorCS2:proNGF:p75NTR complex may further aid in changing the properties of an inactive SorCS2:p75NTR complex or of inactive p75NTR dimers or trimers to trigger signaling. More work will be required to resolve the details of how a ternary SorCS2:proNGF:p75NTR complex is formed and how it triggers signaling. We have extended the discussion to emphasize the different models that have been suggested for proNGF induced p75NTR signaling and have added our SEC-MALS analysis to the manuscript (see also Suppl. Fig. 10).

We have now combined previous Figure 3D and supplementary figure 8 into a new Figure 5 and emphasized with arrows that the orientation of p75NTR (or of NGF depending on the frame of reference) is 180 degrees rotated compared to the TrkA-NGF and the SorCS2-NGF complexes. We now mention in the figure legend that the orientation of p75NTR is upside down in the ternary model.

4. The question of SorCS2 dimerization is approached only at low pH (5.5), using chromatography and light scattering. This should be repeated at neutral pH, as SorCS2 is cell surface exposed and interacts with proneurotrophin ligands at neutral pH.

A: As suggested by the reviewer we have repeated the SEC-MALS analysis of SorCS2 at neutral pH. This shows that also at neutral pH SorCS2 is a dimer. This analysis is added to supplementary figure 3A.

Reviewer #2 (Remarks to the Author):

Review comment:

Leloup et al., "Structure of the SorCS2-proNGF signaling complex reveals a conformation-dependent mechanism regulating neuronal plasticity"

Sortilin and its related proteins are type-1 membrane receptors implicated in neuronal viability and function. Their extracellular domains are characterized by a ten-bladed beta-propeller known as VPS10p domain. Sortilin family proteins are endocytic receptors, which traffic through the trans-Golgi network, endosomal compartments, lysosome, and cell surface. Sortilin family proteins bind to pro-forms of neurotrophic factors (NTs) such as NGF and BDNF as ligands. In addition, the co-receptor p75NTR functions with the Sortilin family proteins to transmit downstream signals into the cytoplasm. Owing to the lack of their complex structures with NTs, their underlying structural mechanisms remain elusive. In this study, Leloup et al. present the crystal structures of the extracellular domain (ECD) of Sortilin-related CNS-expressed receptor 2 (SorCS2) alone and its complex proNGF at 4.0 and 3.6 Å resolutions, respectively. The SorCS2 ECD consists of the VPS10p, two Cys-rich 10CC, and two polycystic kidney disease (PKD), Fibronectin type-III (FN-III), and

unpredicted membrane proximal (SoMP) domains. Both the unliganded and liganded SorCS2 structures exhibit a cross-braced dimer but in different conformations, illustrating the conformational change in the SorCS2 ECD upon binding to proNGF, which seems to be required for binding to the co-receptor p75NTR. The dimeric SorCS2 binds to two proNGF dimers via the VPS10p domain. The vacant side of proNGF can be utilized for the binding to p75NTR. The two PKD domains mediate the dimer interface, which slides upon binding to proNGF.

The structures provide new structural insights into the recognition of NTs by SorCS2 and important clues to understand the structural mechanism for the activation of downstream events including the binding to p75NTR. However, it seems that descriptions on physiological roles and molecular mechanisms of the Sortilin family are insufficient for general readers to understand the importance or interesting points of this study. I would like to suggest the following points to be improved for publication:

1. Introduction, Results, and Discussion sections should be separated. Results section should also be divided into subsections with subheadings to clarify what the authors present.

A: We thank the reviewer for pointing this out. As requested we have extended the introduction, results and discussion and divided the results into sections with subheadings.

2. The current introduction part lacks the details of the functional roles of the Sortilin family proteins in the context of molecular signaling for the regulation of cellular events. How do the Sortilin family proteins regulate synaptic plasticity and cell death/survival? How are neurotrophic ligands, p75NTR, and Trk involved in these events? What is the difference of SorCS1–SorCS3, Sortilin, and SorLA? Why do the Sortilin family proteins bind to proform NTs instead of mature NTs? In addition, the intracellular trafficking of the Sortilin family proteins and its physiological significance should be described. Citation of review article(s) (e.g., Nykjaer and Willnow, Trends in Neuroscience, 35, 261-270 (2012)) may be helpful.

A: The introduction has now been extended to include more background information on the functional roles and structural differences of the VPS10p family as well as the interplay between pro- and mature neurotrophins, VPS10p receptors and Trk/p75NTR.

3. Descriptions of the unliganded and liganded structures should be separated, because these two structures exhibit different conformations. I would like to suggest the authors to make a subsection for each structure.

A: This has been separated as requested.

4. pg 4, How did the authors define the top and bottom of the beta-propeller? Should be explained.

A: To address this comment, we have added the following description at this position in the manuscript: “(by convention the top face is defined as the β -propeller surface that contains the DA and BC loops (Chen et al., 2011)”

5. The liganded structure suggests that the pro region of NGF or BDNF is dispensable for the binding to SorCS2. Comparison between the affinities to the pro-form and mature form of NTs should be required to confirm that the pro region is not actually involved in the binding.

A: We have now performed SPR analysis of NGF and proNGF binding to wild-type and mutant F630N sSorCS2 (see Fig. 4A). This analysis shows both NGF and proNGF interact with SorCS2 and that proNGF interacts with six-fold higher affinity to SorCS2 (Kd of 43 nM) than does mature NGF (Kd of 236 nM). The interaction is abrogated for both NGF and proNGF by the SorCS2 F630N mutation and indicates that the interaction site we observe in the SorCS2-NGF structure is also used by proNGF. This experiment shows that the pro domain itself is also contributing to the interaction with SorCS2 as was shown by others previously (Glerup et al., 2014).

Furthermore, thanks to a comment from reviewer 1 (see comment 2), we have analysed processing of proNGF in the crystallization condition and observe substantial processing of proNGF into NGF over three weeks incubation at room temperature (supplementary figure 11). Consequently, we have rewritten the manuscript considering that the ligand present in the structure is NGF and not proNGF. We believe our previous analyses and conclusions have not changed with the realization that mature NGF instead of proNGF crystallized in complex to SorCS2, since due to absence in the electron density we were not able to comment on any proNGF pro domain contribution.

We have added the new SPR experiments that show both NGF and proNGF bind to sSorCS2 and that the F630 mutant abrogates this interaction. In addition, we have extended the discussion to emphasize that our data does not reveal how the pro domain of NGF contributes to interaction with SorCS2.

6. pg 5, "Biophysical interaction assays ..." may be rewritten as "Surface plasmon resonance analyses ..." to specify the method.

A: This has been done as requested.

7. pg 5, The structure around Phe630 of SorCS2 should be presented to show how the introduced N-linked glycan affects the interface.

A: To address this comment, we have added a panel showing a modeled oligomannose on F630 at the SorCS2-proNGF interface in Fig. 3D, including a zoom-in on the steric clash introduced by the glycan.

8. SEC-MALS and/or SAXS analyses of the SorCS2-proNGF-p75NTR ternary complex may support the proposed structural mechanism of the ternary complex assembly.

A: This experiment was also requested by reviewer 1 (comment 3). We have now performed SEC-MALS experiments of SorCS2:proNGF:p75NTR ectodomains in solution as the reviewer suggests. We used neutral pH and a buffer supplemented with Ca^{2+} as this was shown to be important for ternary complex formation with Sortilin (Feng et al. 2010). Our experiment shows that proNGF and p75NTR interact. SorCS2 however does not interact with either proNGF or p75NTR in this setting as there is no peak shift apparent for SorCS2, nor is there a mass increase. We observe a weight average mass of individual SorCS2 dimers and proNGF:p75NTR complexes that are co-eluting. The proNGF:p75NTR complexes probably exists as a mix of 2:1 and 2:2 complexes similar as observed previously (Feng et al., 2010). The absence of SorCS2:proNGF:p75NTR complexes may be explained by the lack of an avidity effect in solution that would be present on the cell surface if transmembrane proteins SorCS2 and p75NTR bind to proNGF simultaneously. This hypothesis is supported by cell-binding assays that show SorCS2 and p75NTR cooperate in proNGF binding. ProNGF interacts with higher affinity to cells expressing both SorCS2 and p75NTR on the surface compared to cells expressing SorCS2 and p75NTR individually (Glerup et al., 2014).

9. The difference or similarity between SorCS2 and other Sortilin family proteins should be discussed at primary or tertiary structure levels. In addition, discussion regarding future studies and/or applications utilizing the present structures may be helpful for general researchers in biological or medical science fields.

A: We have expanded the introduction to include a section on the similarities and differences between VPS10p members. In addition we have added a section to the results and supplementary figure 4 comparing the structures of VPS10p members now available. We have added a final paragraph to the discussion on future studies and questions that can now be addressed using the new sSorCS2 structures.

10. When resolution limits are defined on the basis of CC(1/2), CC(1/2) in the high resolution shell is typically more than 0.5. I would like to suggest the authors to try structure refinement at lower resolution with CC(1/2) in the high resolution shell being more than 0.5. If such refinement results in worse geometry, the current refinement at 3.6 (liganded) or 4.0 Å (unliganded) resolution may be OK. If not, the effective resolution may be worse than 3.6 or 4.0 Å resolution, because the diffraction data corresponding to high resolution seem to have little structural information.

A: We thank the reviewer for raising this important point. We tested refinement of both structures to data truncated to a resolution where the CC(1/2) was more than 0.5; 4.2 Å for unliganded sSorCS2 and 3.9 Å for the ligand bound sSorCS2. This did not affect the geometry noticeably and we therefore followed the reviewer's advice to adhere to a more conservative data cutoff. We have adjusted the manuscript and the crystallographic table with the updated statistics and deposited the re-refined structures to the protein data bank.

11. pg 17., PDB IDs are missing.

A: The PDB IDs (6FG9 for sSorCS2 and 6FFY for sSorCS2-NGF) have now been added.

12. Figure legends of Supplementary Fig.3, "... indicates a molecular mass of ..." should be "... indicates a molar mass of ...".

A: This has been done as requested.

13. There are several typos (e.g., "... in the in the crystal lattice for ..." in figure legends of Supplementary Fig. 4). Check the manuscript more carefully before resubmission.

A: We thank the reviewer for noticing these typos. We carefully went through the manuscript before resubmission and have hopefully corrected all typos.

14. The PDB validation reports are inconsistent with Table 1. The outlier residues in Ramachandran plot should be corrected unless their unfavorable main-chain conformations are stabilized by some interactions.

A: Table 1 has been corrected so that it is consistent with the validation report. The outlier residues have been fixed and the re-refined models have been submitted to the protein data bank.

Reviewers' comments:

Reviewer #1 (Remarks to the Author):

The revised manuscript by Lecoup and colleagues provides new information that the crystal structure initially described as proNGF: SorCS2 in fact reflects mature NGF: SorCS2 (Supplementary figure 11). The authors also provide new additional gel filtration studies that are unable to document an interaction between proNGF and SorCS2, or a ternary complex of proNGF, SorCS2 and p75, when in solution (Supplementary figure 10). Lastly, new SPR studies (Supplementary figure 10), performed by coupling ligand to the grid, document low affinity (240 nM) interaction of NGF with soluble SorCS2. These SPR studies are largely consistent with prior work by Glerup and colleagues, who documented, when coupling SorCS2 to a grid, 5 nM interaction of proNGF with SorCS2, comparable (low nanomolar) interaction of the prodomain with SorCS2, and negligible interaction of mature NGF with SorCS2.

While this study provides crystallographic data to support the EM imaging which has been published to date on SorCS2, it does not provide insight as to the biological mechanisms which underlie proNGF or SorCS2 function.

There are no known biological consequences of mature NGF-SorCS2 binding, whereas numerous studies describe potent and acute actions of low nanomolar concentrations of proNGF which are mediated by the p75/SorCS2 receptor complex. With the lack of a described functional effect of mature NGF/SorCS2 binding, this manuscript is a long way from a complete mechanistic study that will provide clarity to the field. Based on the very low affinity (this study) or negligible (Glerup et al) interaction of mature NGF with SorCS2, yet the documented high affinity interaction with the prodomain of NGF or proNGF with SorCS, this study is premature, as it fails to provide information about the most relevant interaction of the prodomain region with SorCS2. As such, the abstract is misleading, as it implies that this study "suggests how proNGF simultaneously engages SorCS2 and p75 NTR to trigger signaling" whereas it fails to clarify the most important aspects of the interaction which mediate biological action, and is likely to confuse the field.

Reviewer #2 (Remarks to the Author):

NCOMMS-17-33077-A

Review comment:

Leloup et al., "Structure of the SorCS2-NGF signaling complex provides insights into neuronal plasticity"

The authors have addressed most of my and another reviewer's concerns. However, the following points remain to be improved for publication:

1. The last paragraph of Introduction seems too detailed. It can be described more concisely. In addition, the aim of this study should be described explicitly.
2. In In 154, pg 7, "Supplementary Fig. 8" may be "Supplementary Fig. 4". 10CC-b of SorLA is not shown in Supplementary Fig. 4.
3. In In 170, pg 8, "Surface Plasmon resonance analyses confirm that ..." should be rewritten as "Surface plasmon resonance analyses support the idea that ...".
4. In the SPR analysis of the binding between NGF and SorCS2 (Fig. 4), the baseline increases after every injection. The change in the baseline makes the affinity calculation inaccurate. The analysis of the NGF-SorCS2 binding should be revised. The wash condition after each injection

needs to be improved. Alternatively, addition of BSA (e.g., 0.5%) in the running buffer could inhibit the increase in the baseline in some case.

5. Labeling seems wrong in the figure legends of Supplementary Figure 3.

6. The order of the citation of the figures is not appropriate. The order of the individual panels of the figures should be rearranged to follow the order of their citation in the main text. For example, Fig. 3 should be merged with Fig. 2a,b, whereas Fig. 2c should be separated from Fig. 2.

7. The title seems inappropriate, because this study does not provide any insights into neuronal plasticity. For example, "Structural insights into the complex formation between SorCS2 and NGF" would be appropriate.

8. The manuscript should be thoroughly proofread. There are grammatical, typographical, and stylistic errors.

Reviewers' comments:

Reviewer #1 (Remarks to the Author):

The revised manuscript by Lecoup and colleagues provides new information that the crystal structure initially described as proNGF: SorCS2 in fact reflects mature NGF: SorCS2 (Supplementary figure 11). The authors also provide new additional gel filtration studies that are unable to document an interaction between proNGF and SorCS2, or a ternary complex of proNGF, SorCS2 and p75, when in solution (Supplementary figure 10). Lastly, new SPR studies (Supplementary figure 10), performed by coupling ligand to the grid, document low affinity (240 nM) interaction of NGF with soluble SorCS2. These SPR studies are largely consistent with prior work by Glerup and colleagues, who documented, when coupling SorCS2 to a grid, 5 nM interaction of proNGF with SorCS2, comparable (low nanomolar) interaction of the prodomain with SorCS2, and negligible interaction of mature NGF with SorCS2.

While this study provides crystallographic data to support the EM imaging which has been published to date on SorCS2, it does not provide insight as to the biological mechanisms which underlie proNGF or SorCS2 function.

There are no known biological consequences of mature NGF-SorCS2 binding, whereas numerous studies describe potent and acute actions of low nanomolar concentrations of proNGF which are mediated by the p75/SorCS2 receptor complex. With the lack of a described functional effect of mature NGF/SorCS2 binding, this manuscript is a long way from a complete mechanistic study that will provide clarity to the field. Based on the very low affinity (this study) or negligible (Glerup et al) interaction of mature NGF with SorCS2, yet the documented high affinity interaction with the prodomain of NGF or proNGF with SorCS, this study is premature, as it fails to provide information about the most relevant interaction of the prodomain region with SorCS2. As such, the abstract is misleading, as it implies that this study “suggests how proNGF simultaneously engages SorCS2 and p75 NTR to trigger signaling” whereas it fails to clarify the most important aspects of the interaction which mediate biological action, and is likely to confuse the field.

A: We thank the reviewer for pointing out that the abstract may be confusing the field. To address this we have adjusted the manuscript and replaced the last sentence of the abstract with: “Taken together, our data reveal a structurally flexible SorCS2 receptor that employs the large β -propeller as a ligand binding platform”. Now we keep more in line with our own data and this should prevent any confusion.

We agree with the reviewer on the importance that the pro domain of NGF plays in binding to SorCS2 and we discuss this in the manuscript. Furthermore, for the first time we quantify the difference in affinity between proNGF and NGF for SorCS2, and show that binding of proNGF is six fold stronger. Binding of NGF directly to SorCS2 has been shown by Glerup et al. (Glerup et al., 2014) although no quantification of the NGF - SorCS2 interaction is reported. A quantified interaction of 40 nM has been reported previously for NGF binding to SorCS3 (a SorCS2 paralog with 47% sequence identity to SorCS2) and of 70 nM for proNGF binding to SorCS3 (Westergaard et al., 2005). We believe our data and that of others do show that NGF interacts with SorCS2, albeit with reduced affinity compared to proNGF. We have now indicated the proNGF pro domain with dotted lines in

the left panel of Fig. 6d to emphasise its importance in signalling and its unknown position. While we do not reveal how the pro domain of proNGF binds to SorCS2 it will be very difficult to do so due to its unstructured/disordered nature.

Our structural data indicate an NGF binding interface on SorCS2 that appears to be functionally conserved for proNGF binding as is also supported by our SPR experiments that show binding of proNGF to SorCS2 is disrupted by an NGF binding-based mutation on SorCS2. As such our SorCS2-NGF crystal structure albeit incomplete to describe proneurotrophin-SorCS2 binding, will be valuable to the broader research community.

Reviewer #2 (Remarks to the Author):

NCOMMS-17-33077-A

Review comment:

Leloup et al., "Structure of the SorCS2-NGF signaling complex provides insights into neuronal plasticity"

The authors have addressed most of my and another reviewer's concerns. However, the following points remain to be improved for publication:

1. The last paragraph of Introduction seems too detailed. It can be described more concisely. In addition, the aim of this study should be described explicitly.

A: We have shortened the last paragraph of the introduction and added the aim of this study.

2. In In 154, pg 7, "Supplementary Fig. 8" may be "Supplementary Fig. 4". 10CC-b of SorLA is not shown in Supplementary Fig. 4.

A: We corrected the reference to Supplementary Fig. 4. We have added an additional panel (d) to Supplementary Fig. 4 that now shows 10CC-b of SorLA together with 10CC-b of sortilin and SorCS2.

3. In In 170, pg 8, "Surface Plasmon resonance analyses confirm that ..." should be rewritten as "Surface plasmon resonance analyses support the idea that ...".

A: This has been rewritten as suggested.

4. In the SPR analysis of the binding between NGF and SorCS2 (Fig. 4), the baseline increases after every injection. The change in the baseline makes the affinity calculation inaccurate. The analysis of the NGF-SorCS2 binding should be revised. The wash condition after each injection needs to be improved. Alternatively, addition of BSA (e.g., 0.5%) in the running buffer could inhibit the increase in the baseline in some case.

A: We employed a nonregeneration protocol to analyse the NGF-SorCS2 interaction as regeneration was incomplete. This is a well-documented method (see for example Tang et al., 2006) and should not affect the affinity calculation as long as the relative amount of nonregenerated analyte is negligible compared to the free analyte. This is the case in our experiment. To emphasise that we used the nonregeneration method we have added to the legend of figure 4 "Incomplete

regeneration for NGF was accounted for by treating the data as an equilibrium titration experiment (Tang et al., 2006)” and to the methods section “Incomplete regeneration of NGF was accounted for by zeroing the data only at the start of the injection series (Tang et al., 2006)”.

5. Labeling seems wrong in the figure legends of Supplementary Figure 3.

A: We thank the reviewer for pointing this out and have corrected this.

6. The order of the citation of the figures is not appropriate. The order of the individual panels of the figures should be rearranged to follow the order of their citation in the main text. For example, Fig. 3 should be merged with Fig. 2a,b, whereas Fig. 2c should be separated from Fig. 2.

A: To correct the order, we have moved Fig. 2c as a new figure to figure 5. Fig 2a,b are now followed by Fig. 3. Figures have been renumbered in the manuscript accordingly.

7. The title seems inappropriate, because this study does not provide any insights into neuronal plasticity. For example, “Structural insights into the complex formation between SorCS2 and NGF” would be appropriate.

A: To address this we have now renamed the title into “Structural insights into SorCS2 – Nerve Growth Factor complex formation”. We thank the referee for suggesting this.

8. The manuscript should be thoroughly proofread. There are grammatical, typographical, and stylistic errors.

A: We thank the reviewer for pointing this out. We have corrected grammatical, typographical and stylistic errors throughout the manuscript.